# Evaluation of SARS-CoV-2 entry, inflammation and new therapeutics in human lung tissue cells

Judith Grau-Expósito[1,☯], David Perea[1,☯], Marina Suppi[1], Núria Massana[1], Ander Vergara[2], Maria José Soler[2], Benjamin Trinite[3], Julià Blanco[3,4], Javier García-Pérez[5,6], José Alcamí[5,6,7], Anna Serrano-Mollar[8,9], Joel Rosado[10], Vicenç Falcó[1], Meritxell Genescà[1]*, Maria J. Buzon[1]*

1 Infectious Diseases Department, Vall d'Hebron Research Institute (VHIR), Hospital Universitari Vall d'Hebron, Universitat Autònoma de Barcelona, VHIR Task Force COVID-19, Barcelona, Spain, 2 Nephrology Research Department, Vall d'Hebron Research Institute (VHIR), Hospital Universitari Vall d'Hebron, Universitat Autònoma de Barcelona, VHIR Task Force COVID-19, Barcelona, Spain, 3 IrsiCaixa AIDS Research Institute, Germans Trias i Pujol Research Institute (IGTP), Can Ruti Campus, Autonomous University of Barcelona (UAB), Badalona, Spain, 4 University of Vic–Central University of Catalonia (UVic-UCC), Vic, Spain, 5 AIDS Immunopathology Unit, National Center of Microbiology, Instituto de Salud Carlos III, Madrid, Spain, 6 Centro de Investigación Biomédica en Red de Enfermedades Infecciosas, Instituto de Salud Carlos III (ISCIII), Madrid, Spain, 7 Clinic HIV Unit, Hospital Clinic, IDIBAPS, Barcelona, Spain, 8 Experimental Pathology Department, Institut d'Investigacions Biomèdiques de Barcelona, Consejo Superior de Investigaciones Científicas (IIBB-CSIC), Institut d'Investigacions Biomèdiques August Pi i Sunyer (IDIBAPS), Barcelona, Spain, 9 Centro de Investigaciones Biomédicas en Red de Enfermedades Respiratorias (CIBERES), Madrid, Spain, 10 Thoracic Surgery and Lung Transplantation Department, Vall d'Hebron Institut de Recerca (VHIR), Hospital Universitari Vall d'Hebron, VHIR Task Force COVID-19, Barcelona, Spain

☯ These authors contributed equally to this work.
* meritxell.genesca@vhir.org (MG); mariajose.buzon@vhir.org (MJB)

**Data Availability Statement:** All relevant data are within the manuscript and its Supporting Information files.

## Abstract

The development of physiological models that reproduce SARS-CoV-2 infection in primary human cells will be instrumental to identify host-pathogen interactions and potential therapeutics. Here, using cell suspensions directly from primary human lung tissues (HLT), we have developed a rapid platform for the identification of viral targets and the expression of viral entry factors, as well as for the screening of viral entry inhibitors and anti-inflammatory compounds. The direct use of HLT cells, without long-term cell culture and in vitro differentiation approaches, preserves main immune and structural cell populations, including the most susceptible cell targets for SARS-CoV-2; alveolar type II (AT-II) cells, while maintaining the expression of proteins involved in viral infection, such as ACE2, TMPRSS2, CD147 and AXL. Further, antiviral testing of 39 drug candidates reveals a highly reproducible method, suitable for different SARS-CoV-2 variants, and provides the identification of new compounds missed by conventional systems, such as VeroE6. Using this method, we also show that interferons do not modulate ACE2 expression, and that stimulation of local inflammatory responses can be modulated by different compounds with antiviral activity. Overall, we present a relevant and rapid method for the study of SARS-CoV-2.

**Funding:** This work was primarily supported by a grant from the Health Department of the Government of Catalonia (DGRIS 1_5) to M.G and MJ.B. This work was additionally supported in part by the Spanish Health Institute Carlos III (ISCIII, PI17/01470 to M.G; PI19CIII/00004 to J.A; PI21CIII/00025 to J.G-P and COV20-00679 (MPY 222-20) to J.G-P), the Spanish Secretariat of Science and Innovation and FEDER funds (grant RTI2018-101082-B-I00 [MINECO/FEDER]) to MJ. B, the Spanish AIDS network Red Temática Cooperativa de Investigación en SIDA and the European Regional Development Fund (ERDF) (RD16/0025/0007 to MJ.B and RD16CIII/0002/0001 to J.A), the Fundació La Marató TV3 (grants 201805-10FMTV3 and 202104FMTV3 to MJ.B; 201814-10FMTV3 and 202112FMTV3 to M.G), the Gilead fellowships GLD19/00084 to M.G and GLD18/00008 to MJ.B and the Becas Taller Argal 2020 to MJ.B. Salary for MJ.B is supported by the Miguel Servet program funded by the Spanish Health Institute Carlos III (CP17/00179). N.M and D.P are supported by a Ph.D. fellowship from the Vall d'Hebron Institut de Recerca (VHIR).The funders had no role in study design, data collection and analysis, the decision to publish, or preparation of the manuscript.

**Competing interests:** The authors have declared that no competing interests exist.

## Author summary

The early stages of laboratory identification of therapeutics against pathogens is usually based on the use of immortalized cell lines, as exemplified by many studies screening anti-virals against SARS-CoV-2. Cell lines are manipulated for their continuous growth which offers several advantages, however they do not fully reproduce the behavior of primary cells nor the complexity of heterogeneous populations. In this study, we overcome this limitation by using surgical resections to establish human lung tissue (HLT) cell cultures ready for drug evaluation. First, we show that HLT preserves lung cell composition, including the main SARS-CoV-2 cellular target, namely alveolar type-2 cells, as well as the proteins required for viral entry into the cells: ACE2, CD147, TMPRSS2 and AXL. Moreover, using HLT cells we identified new antiviral drug candidates missed by conventional systems, and anti-inflammatory compounds that module molecules associated with SARS-CoV-2 infection. In summary, we have established a physiological model that can be used for the identification of novel anti-SARS-CoV-2 therapeutics and other respiratory pathogens.

## Introduction

Only one antiviral against SARS-CoV-2, remdesivir, has been approved for the treatment of COVID-19 in adults and pediatric patients (12 years of age and older) requiring hospitalization [1,2]. Moreover, molnupiravir, a ribonucleoside analog that inhibits SARS-CoV-2 replication, [3,4], and paxlovid, an inhibitor of the Mpro viral protease [5] have showed significant potency in animal models. Human clinical trials in non-hospitalized patients are ongoing (NCT04575597 and NCT04960202), which could translate into the first oral antiviral medicines approved for COVID-19. Further, the development of *Acute Respiratory Distress Syndrome* in severe COVID-19 patients has been linked to dysregulated inflammatory responses. In this regard, treatment with the glucocorticoid dexamethasone decreased 28-day mortality among patients receiving invasive mechanical ventilation, but little benefit was observed in patients without respiratory support [6]. Despite these major advances in treatment options for COVID-19, the rapid identification of new antivirals that could be easily transferred into the clinic is still of paramount importance, particularly with the potential emergence of drug-resistant variants.

Screening of novel drug candidates is often performed using cell lines. In this sense, the most widely used cell lines for SARS-CoV-2 studies are epithelial cells derived either from lung (Calu-3), kidney (VeroE6), or colon (CaCo-2) [7]. These immortalized systems are highly reproducible and easy to handle but lack physiological relevance. The differential gene expression profiling of cell lines compared with primary cells from tissues might significantly affect important enzymes involved in the viral replication cycle. For instance, the level of ACE2 expression, the main receptor used by SARS-CoV-2 for viral entry, is variable among several cell lines [8], while only a small fraction of alveolar type II (AT-II) cells, the main target for SARS-CoV-2 in the distal lung, express ACE2 [9,10]. In addition, SARS-CoV-2 spike (S) glycoprotein, which is responsible for viral entry into target cells, can be activated by several host proteases, such as furin, transmembrane serine proteinase 2 (TMPRSS2) and cathepsin L, in a pH-dependent or independent manner [11,12]. Whereas in some cell lines S protein is activated by endosomal pH-dependent protease cathepsin L, in airway epithelial cells viral entry depends on the pH-independent TMPRSS2 protease [12]. Thus, it is currently not well defined

if SARS-CoV-2 may utilize multiple cell-type-specific host proteins for viral replication in primary target tissues and therefore, the potency of therapeutics directed against these proteins may also differ.

Further, inflammatory immune responses might also impact viral dynamics in the lung by affecting the expression of entry receptors. In this sense, early studies discovered that ACE2 was a human interferon-stimulated gene (ISG); IFN-β and IFN-γ were shown to strongly upregulate the expression of ACE2 at the mRNA and cell surface protein levels, indicating that inflammatory molecules could shape cell susceptibility to viral infection [13]. However, how anti-inflammatory drugs may affect ACE2 expression and facilitate SARS-CoV-2 infection remains to be elucidated. One study reported that the use of nonsteroidal anti-inflammatory drugs (NSAIDs), such as ibuprofen, was linked to enhanced ACE2 expression in a diabetic-induced rat model [14] and other reports raised alarms regarding the possible role of NSAIDs at increasing susceptibility to SARS-CoV-2 infection [15,16]. On the contrary, experimental and clinical evidence showed that medium-to-low-dose glucocorticoids may play a protective role in COVID-19 by activating ACE2 and suppressing the associated cytokine storm [17]. Overall, the use of more relevant and physiological models for the study of SARS-CoV-2 infection, the identification of drug candidates, or the impact of new therapeutics on the disease, could significantly advance the successful translation of the results into the clinic.

Primary epithelial cell cultures of nasal and proximal airway epithelium have been used to study SARS-CoV-2 infection in the upper airways [10,18–20]. Similarly, organ on-chip and organoid models of AT-II cells have been successfully developed [19,21]. While very useful, these models require long-term culture (sometimes several weeks) combined with the addition of cytokines that might change cell functionality [22]. The direct use of human lung tissue (HLT) cells offers important advantages over other *in vitro* and *in vivo* approaches for several reasons; it mimics the main site of viral replication in the lung, contains all heterogeneous cell components present in the tissue (with greater functional complexity compared to cell lines), and the cells are not subjected to long-term culture nor exposed to *in vitro* differentiation approaches. In the past, similar lung models have been successfully established to study the effect of allergens and inflammatory stimuli [23,24]. Importantly, using HLT cells allow mimicking an inflammatory local response that could be attenuated by anti-inflammatory drugs, providing a low/medium throughput screening of anti-inflammatory candidates for the treatment of airway diseases [25]. Significantly, lung tissues not only can be infected with SARS-CoV-2, but also generate local immune responses to viral infection [26]. Considering all these factors, here we aimed to characterize a physiological human lung tissue system, which could be used for the study of virus-host interactions and the identification of potential antiviral compounds and their capacity to modify local inflammation. A graphical overview of our approximation is illustrated in **S1 Fig**.

## Results

### Characterization of HLT cells

Non-neoplastic lung parenchyma was obtained from hospitalized non-COVID-19 patients undergoing thoracic surgery. First, we optimized cell culture and digestion conditions, since the methodology used for tissue processing can significantly impact cell-type yield, viability and function of target cell populations. We focused on the preservation of EpCAM+ cells expressing HLA-DR [27,28], which in adult alveolar parenchyma is characteristic of AT-II cells (thereafter referred as enriched AT-II cells). We also aimed to preserve several hematopoietic cell subsets, as shown in the flow cytometry gating strategy (**S2A Fig**). We observed that collagenase outperformed liberase and trypsin at preserving the enriched population of

AT-II cells, the main SARS-CoV-2 target (**S3A and S3B Fig**). Among the hematopoietic cells present in lung parenchyma, we identified CD3 T lymphocytes (which represented 7.39% ± 6.97 out of the total living cells), myeloid dendritic cells (0.10% ± 0.06), monocytes/macrophages subsets (1.36% ± 1.27), neutrophils (5.00% ±4.28) and alveolar macrophages (0.21% ± 0.16). Moreover, out of the non-hematopoietic cells (CD45-), enriched AT-II and endothelial cells represented 1.11% ± 1.16 and 0.78% ± 0.81 of the total living cells, respectively. Based on EpCAM expression and lack of CD31 and CD45, other epithelial cells including AT-I cells represented 0.52% ± 0.46 out of the living fraction (**Fig 1A**). All these populations have been previously identified in human lungs [26,29,30]. Of note, other abundant structural cell subsets were not defined by specific phenotypic markers within the same panel. Moreover, the nature of AT-II cells in lung cell suspensions was also addressed by staining with phosphatase alkaline (**Fig 1B**), which is expressed in AT-II cells both *in vitro* and *in vivo* [31], and by detection of Surfactant Protein C, which is expressed exclusively by fully differentiated AT-II cells [32,33] (**S3C Fig**).

Next, we focus on the expression of previously identified proteins involved in viral entry. Single-cell transcriptome studies have shown that ACE2, one of the main host cell surface receptors for SARS-CoV-2 attachment and infection, is predominantly expressed by AT-II cells [10,34]. Moreover, ACE2 expression wanes in distal bronchiolar and alveolar regions paralleled by SARS-CoV-2 infectivity [10]. In human lung parenchyma-derived cells, we found ACE2 expression mainly associated to the population enriched in AT-II cells (**Figs 1C and S3D**), a finding that was confirmed by immunohistochemistry in concomitant tissue samples (**Fig 1D**). The percentage of cells expressing ACE2 was rather small and varied between individuals (6.23% ± 3.47) (**Fig 1C**), as previously described [9]. We also studied the expression of CD147, which has been reported as a route of SARS-CoV-2 infection in *in vitro* and *in vivo* models [35]. CD147 was ubiquitously expressed in several hematopoietic and non-hematopoietic cells (**Fig 1C**). Importantly, 92.3% ± 2.4 of the population enriched in AT-II cells expressed CD147 (**Fig 1C**). Similarly, and in agreement with other studies [36,37], TMPRSS2 protease expression was also identified in several subsets, including enriched AT-II cells (31.08% ± 7.06) (**Fig 1C**). When we studied the double expression of ACE2 and TMPRSS2, or ACE2 and CD147, we found that only 1.31% ± 0.70 and 3.02% ± 1.84 of AT-II cells expressed both markers, respectively (**S4A Fig**). Last, we studied the expression of AXL, another candidate receptor for SARS-CoV-2 entry in lung cells [38]. We detected high expression of AXL on myeloid cells, while only a fraction of the enriched AT-II cells (mean of 3.82% ± 4.71) expressed AXL (**S4B Fig**). Overall, we found that human lung cell suspensions preserved critical populations and factors required for SARS-CoV-2 infection.

## Susceptibility of HLT cells to SARS-CoV-2 viral entry

Next, we assessed if HLT cells were susceptible to viral infection. We generated pseudotyped vesicular stomatitis virus (VSV) viral particles bearing the D614G form of the S protein of SARS-CoV-2 and expressing either luciferase (VSV*ΔG (Luc)-S) or GFP (VSV*ΔG (GFP)-S) reporter genes upon cell entry. As a control, we used the VSV-G virus, which has very broad cell tropism. As expected, VeroE6 cells were highly susceptible to SARS-CoV-2 entry, as demonstrated for VSV*ΔG (GFP)-S and VSV-ΔG (Luc)-S (**Fig 2A**). Of note, camostat, an inhibitor of the host protease TMPRSS2, did not inhibit cell entry in this cell line, consistent with its lack of expression of this protein (**Fig 2B**). Anti-ACE2 antibody blocked more than 90% of VSV*ΔG (Luc)-S, yet was inactive for VSV-G (Luc) infection (**Fig 2B**). This observation has been widely reported before [12,39,40], and identifies ACE2 as the main cell receptor required for viral entry in VeroE6. Importantly for viral pathogenesis, it has been postulated that

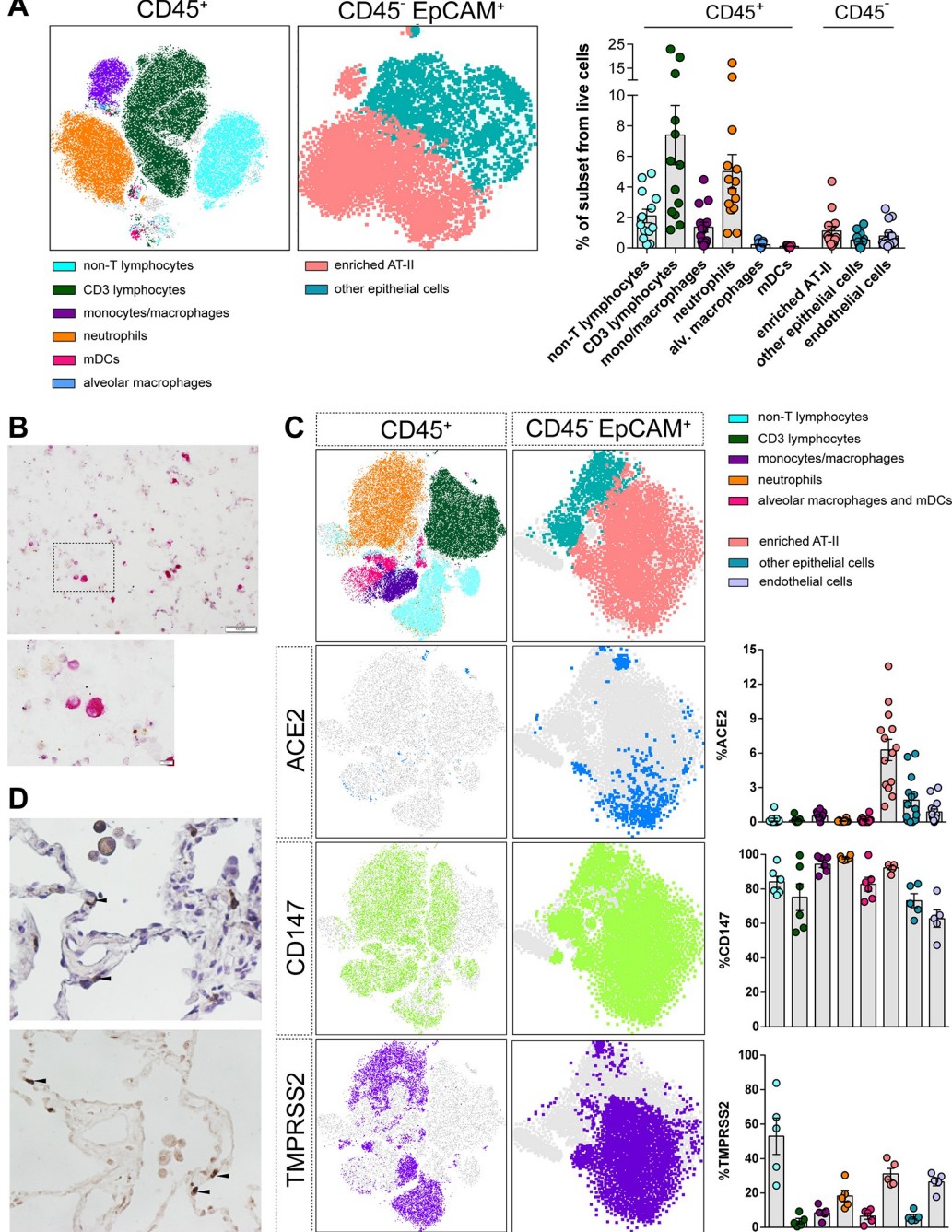

**Fig 1. Phenotyping of human lung cells.** (**A**). *t*-distributed Stochastic Neighbor Embedding (t-SNE) representation displaying the major cell clusters present in the CD45+ and CD45- EpCAM+ fractions of a representative human lung tissue. The vertical bars in the right panel show the frequency of each subset relative to live cells. All cell subsets were identified as shown in S2A Fig. mDCs, myeloid dendritic cells; enriched AT-II, enriched fraction in alveolar type 2. (**B**). Phosphatase alkaline positive AT-II cells (pink staining) were detected in a cytospin obtained from human lung tissue cells and observed at 10x. Lower panel shows a high magnification (40x) of the black square. Scale bars are 100 μm and 10 μm in top and bottom panels, respectively. (**C**). *t*-distributed Stochastic Neighbor Embedding (tSNE) representation for ACE2, CD147 and TMPRSS2 expression in CD45+ and CD45-EpCAM+ fractions from a representative lung tissue. Right graphs show the percentage of expression of each entry factor in the different cell subpopulations, which were identified as in Fig 1A with some modifications for the identification of myeloid cells and neutrophils (From big cells: monocytes/macrophages, CD11c+HLA-DR+ CD14+; Alveolar macrophages and mDCs, CD11c+ HLA-DR+ CD14-; Neutrophils, CD11c- HLA-DR- CD14- CD3-). (**D**). Images of ACE2 immunohistochemical staining in human lung tissue sections at 40x magnification, counterstained with haematoxylin (top) or without (bottom). Black arrows indicate staining of ACE2 in AT-II cells (upper panel). Mean±SEM is shown for all graphs.

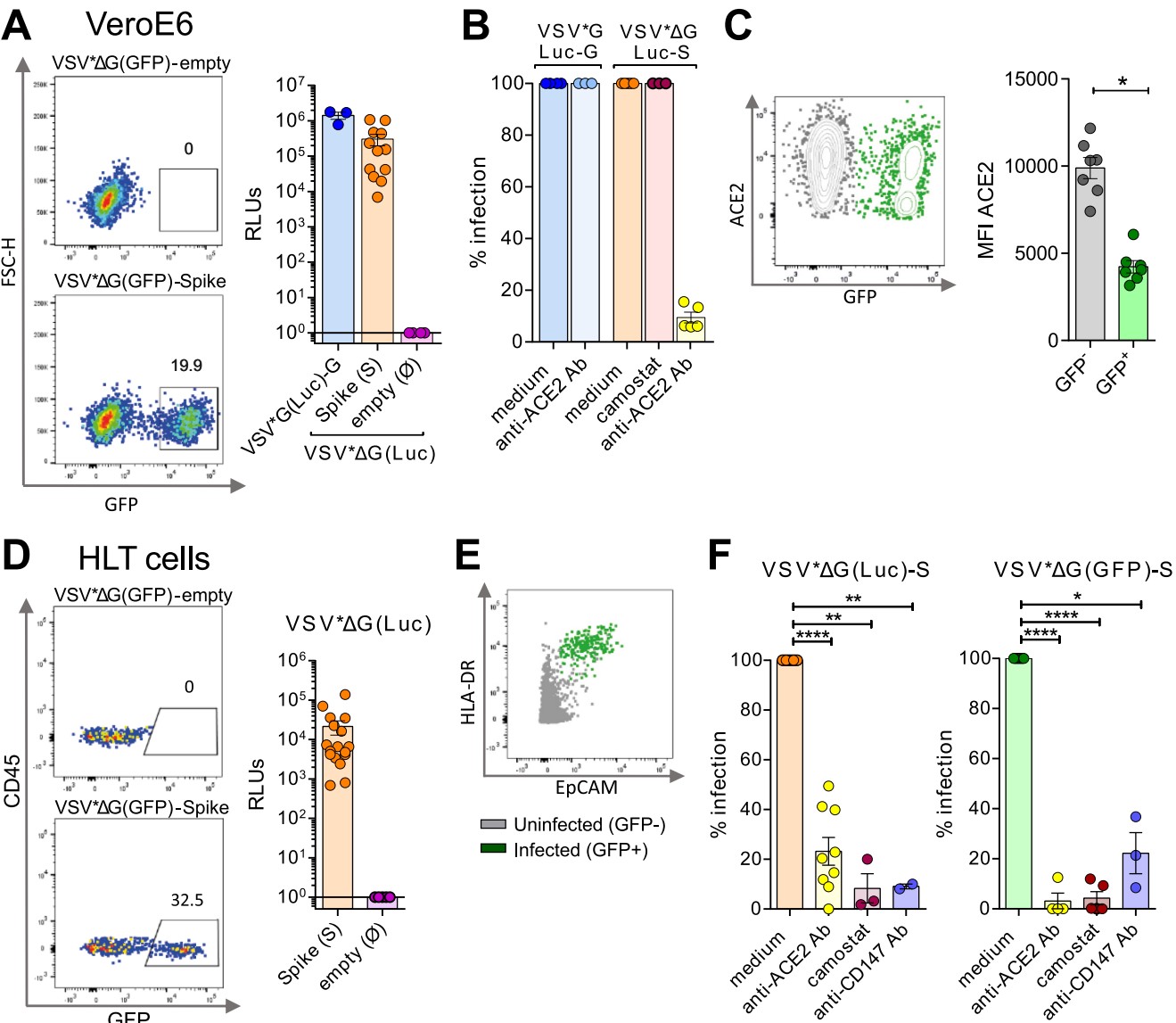

**Fig 2. Susceptibility of VeroE6 and the HLT model to SARS-CoV-2 viral entry.** VeroE6 and HLT cells were infected with two different viral constructs (GFP and Luciferase) expressing the spike protein upon viral entry; VSV*ΔG(GFP)-Spike and VSV*ΔG(Luc)-Spike. (**A**) Representative flow cytometry plots of VeroE6 cells infected with VSV*ΔG(GFP)-Spike or the background form VSV*ΔG(GFP)-empty (left panel); and luciferase activity (RLUs; relative light units) at 20h post-infection with the pseudotyped VSV*G(Luc)-G, the VSV*ΔG(Luc)-Spike or the background form VSV*ΔG(Luc)-empty (right panel). (**B**) Percentage of viral entry after treatment with anti-ACE2 antibody (10μg/ml) and camostat (100μM) in VeroE6 cells infected with the pseudotyped virus expressing the control G protein or the spike from SARS-CoV-2. (**C**) A flow cytometry plot showing ACE2 expression in GFP$^+$ VeroE6 cells. Right graph shows mean fluorescence intensity (MFI) of ACE2 in both infected and uninfected fractions, based on GFP expression. (**D**) Representative flow cytometry plots of HLT cells infected with the viral construct expressing the spike protein (VSV*ΔG(GFP)-Spike) or the background form (VSV*ΔG(GFP)-empty) (left panel); and luciferase activity (RLUs; relative light units) at 20h post-infection with the VSV*ΔG(Luc)-Spike or the background form VSV*ΔG(Luc)-empty (right panel). Infection was measured as the percentage of GFP or RLUs, respectively. (**E**) Susceptible HLT cells to viral entry (identified as GFP$^+$ cells) compatible with an AT-II phenotype, determined by the co-expression of HLA-DR and EpCAM in the CD45$^-$CD31$^-$ fraction of live cells. (**F**) Bar plots showing the percentage of viral entry inhibition on HLT cells in the presence of anti-ACE2 antibody (15μg/ml), camostat (100μM) or anti-CD147 antibody (25μg/ml) after cell challenge with VSV*ΔG(Luc)-Spike (left graph) or VSV*ΔG(GFP)-Spike (right graph). Mean±SEM is shown for all graphs. Data in panel 2C were analyzed by a Wilcoxon matched-pairs test; *p<0,05. Data in panel 2F were analyzed by one sample t-test; *p<0.05, **p<0.01, **p<0.0001.

SARS-CoV-2 S protein might downregulate ACE2 expression, as previously observed for SARS-CoV [41]. We consistently observed a significant strong reduction in ACE2 expression after viral entry (**Fig 2C**).

We then evaluated the susceptibility of HLT cells to viral entry, using the same viral constructs. HLT cells were readily infected with pseudotyped S particles (VSV*ΔG (Luc)-S and VSV*ΔG (GFP)-S), with the natural donor variation representative of primary samples (**Fig 2D**). As expected [10], lung cells enriched with the AT-II phenotype were identified as the main SARS-CoV-2 cell targets in steady conditions (**Fig 2E**). Blockade of ACE2 resulted in a donor-dependent reduction of viral infectivity, ranging from 50 to 100% (**Fig 2F**). Camostat significantly inhibited viral entry in all HLT assays, although the entry process was not always completely abrogated (**Fig 2F**), suggesting that AT-II cells may become infected through the use of alternative factors [42]. Similarly, the presence of an anti-CD147 antibody and the recombinant protein AXL inhibited SARS-CoV-2 entry (**Figs 2F and S4C**). Collectively, these data indicate that HLT cells are susceptible to SARS-CoV-2 viral entry, and that ACE2, CD147, TMPRSS2 and AXL are important proteins required for viral entry in human lung cells. Thus, these results support the value of the direct use of HLT cells to successfully study SARS-CoV-2 viral entry, and related mechanisms, in a more physiological system compared to immortalized cell lines.

## Antiviral assays in HLT cells

To validate the HLT system as a platform for the rapid screening of antiviral candidates, we assayed potential antiviral compounds, most of them previously identified by computational methods with predicted ability to inhibit SARS-CoV-2 cell entry due to their interaction with S protein or with the interface S-ACE2 [43]. A detailed description of the 39 selected drugs is available in **S1 Text**. HLT cells were exposed to VSV*ΔG (Luc)-S virus in the presence of 1/5 serial dilution of the different tested compounds. 20h post-exposure, antiviral activity and cell viability were measured by luminescence. Cell viability for the individual HLT populations, before and after SARS-CoV-2 infection, was measured by flow cytometry and is shown in **S4D Fig**. Antiviral results in HLT cells were systematically compared with parallel testing in the cell line VeroE6. Among the 39 drugs that were evaluated in our study, 15 of them (38%) showed some antiviral activity against SARS-CoV-2 with $EC_{50}$ values ranging from 0.37μM to 90μM (**S2 Text**). From these, 25% had concordant results between both models (**Fig 3**). Cepharanthine, a naturally occurring alkaloid reported to have potent anti-inflammatory and antiviral properties, was one of the most potent antivirals identified in both systems, with $EC_{50}$ of 0.46μM and 6.08μM in VeroE6 and HLT cells, respectively (**Figs 3A and 3B**). Of note, we observed cell toxicity at the highest concentrations ($CC_{50\ VeroE6} = 22.3$μM; $CC_{50\ HLT} = 13.8$μM), which translated in satisfactory *selectivity indexes* (SI = $CC_{50}$ /$EC_{50}$ of $SI_{VeroE6} = 48.47$ and $SI_{HLT} = 2.64$) [44]. The anti-SARS-CoV-2 activity of hydroxychloroquine, a compound known to interfere with endosomal acidification, which is necessary for cathepsin activity, has been extensively reported [45,46]. In our study, we observed that hydroxychloroquine was equally effective at inhibiting viral entry in VeroE6 and HLT cells ($EC_{50\ VeroE6} = 1.58$μM; $EC_{50\ HLT} = 3.22$μM) (**Fig 3A and 3B**), with no apparent cytotoxicity (**Fig 3C**). Ergoloid, an approved drug used for dementia, and recently identified as a potential inhibitor of main protease of SARS-CoV-2 [47], induced ~90% of viral entry inhibition in HLT cells at non-toxic concentrations (**Fig 3A–3C**). Indeed, *SI* for this compound was higher in the HLT model than in VeroE6 cells ($SI_{VeroE6} = 3.9$; $SI_{HLT} = 11.38$). Similarly, ivermectin, a broad-spectrum antiparasitic compound, showed very similar antiviral potency in both models, however *SI* greatly differed between them ($SI_{VeroE6} = 1.4$; $SI_{HLT} = 7.75$). Additionally, we plot the individual $EC_{50}$

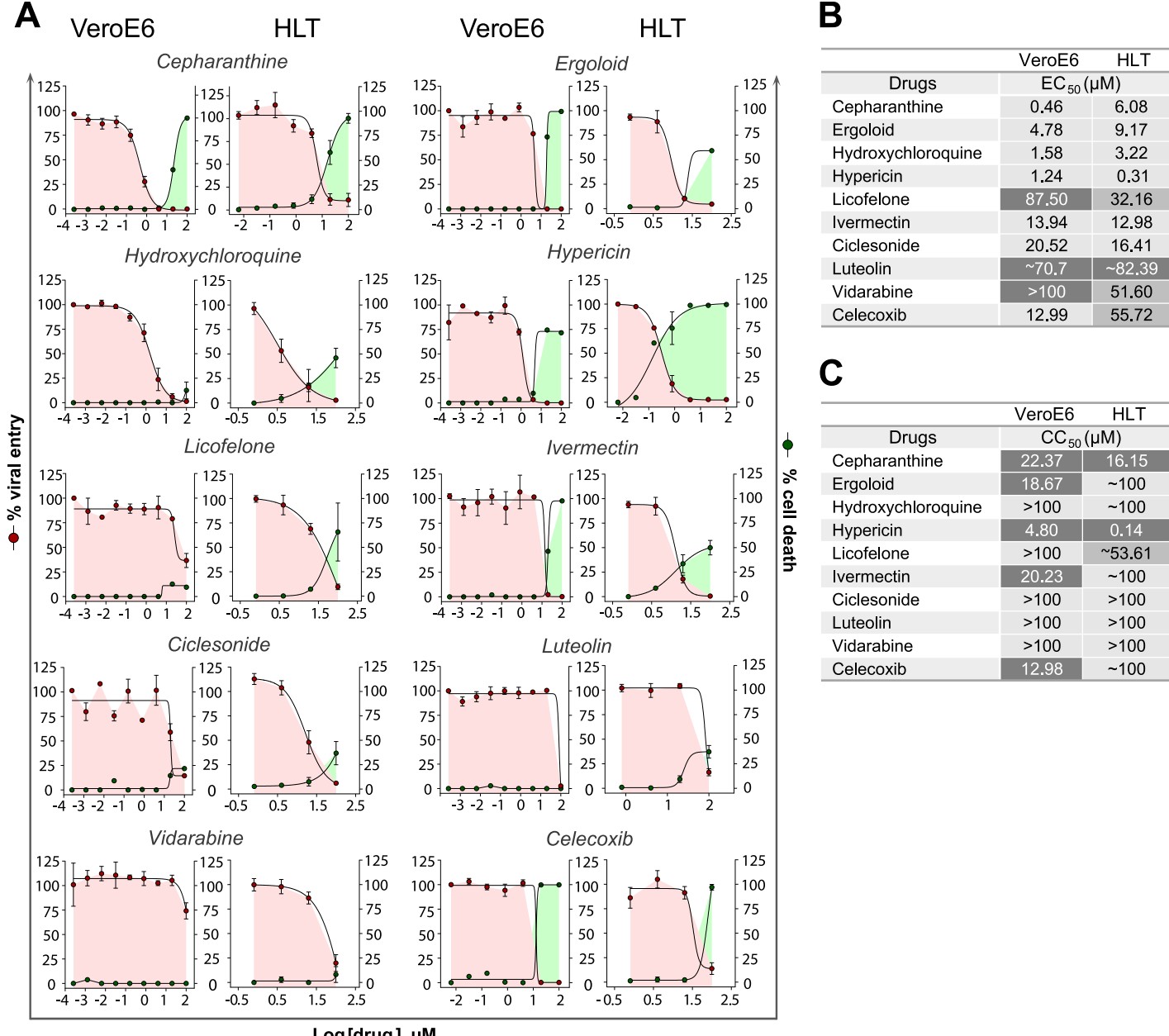

**Fig 3. Antiviral assays with concordant results between models.** (**A**). Percentage of viral entry in VeroE6 and HLT cells exposed to VSV*ΔG(Luc)-Spike in the presence of cepharanthine, ergoloid, hydroxychloroquine, hypericin, licofelone, ivermectin, ciclesonide, luteolin, vidarabine and celecoxib. Drugs were used at concentrations ranging from 100μM to 0.256nM, in VeroE6, and to 0.8μM in lung cells. Non-linear fit model with variable response curve from at least three independent experiments in replicates is shown (red lines). Cytotoxic effect on VeroE6 cells and HLT exposed to drug concentrations in the absence of virus is also shown (green lines). (**B**). EC$_{50}$ values of each drug in VeroE6 and HLT cells. (**C**). CC$_{50}$ values of each drug are shown for VeroE6 and for HLT cells.

values obtained from the different donors. We show that the assay was reproducible (**S4E Fig**), highlighting the suitability of the HLT system for the rapid identification of antivirals.

We also detected discordant antiviral results between both models (**Fig 4**). Four drugs inhibited SARS-CoV-2 entry in HLT cells without affecting cell viability, with no antiviral effect in VeroE6 (**Fig 4A–4C**). As expected and mentioned before, camostat, a TMPRSS2 inhibitor [48], was not active in VeroE6 cells due to the lack of TMPRSS2 expression in this

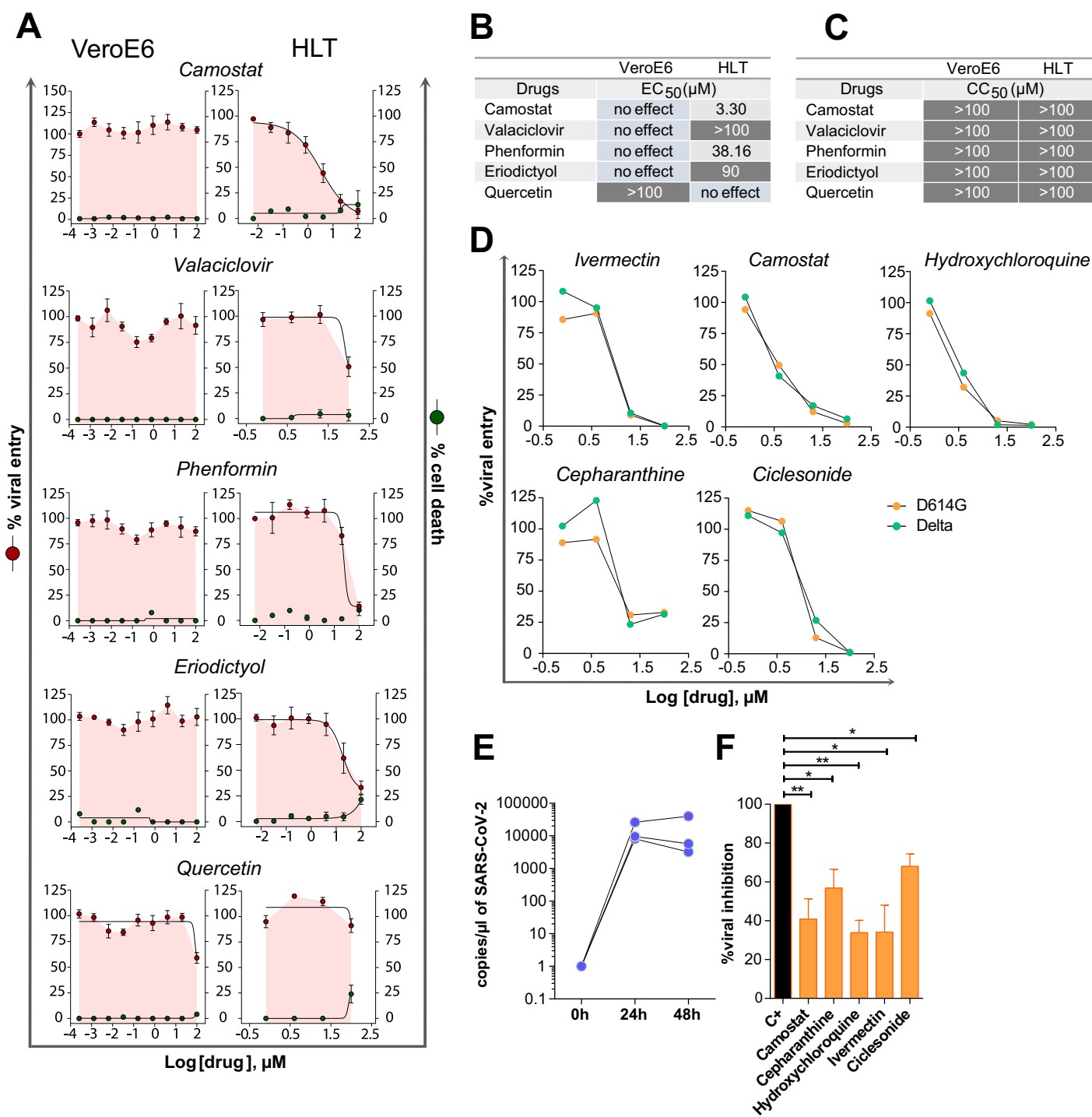

**Fig 4. Antiviral assays with discordant results between models.** (**A**). Percentage of viral entry in VeroE6 and HLT cells exposed to VSV*ΔG(Luc)-Spike in the presence of camostat, valaciclovir, phenformin, eriodictyol and quercetin. Drugs were used at concentrations ranging from 100μM to 0.256 nM, in VeroE6, and to 0.8μM in lung cells. Non-linear fit model with variable response curve from at least three independent experiments in replicates is shown (red lines). Cytotoxic effect on VeroE6 cells and HLT exposed to drug concentrations in the absence of virus is also shown (green lines). (**B**). $EC_{50}$ values of each drug in VeroE6 and HLT cells. (**C**). $CC_{50}$ values of each drug are shown for VeroE6 and HLT cells. (**D**). Percentage of viral entry in HLT cells exposed to VSV*ΔG(Luc)-Spike-delta and VSV*ΔG(Luc)-Spike-D614G variants in the presence of ivermectin, camostat, hydroxychloroquine, cepharanthine and ciclesonide. Drugs were used at concentrations ranging from 100μM to 0.8μM. (**E**) Number of viral genomes/μl at 24h and 48h after infection with replication-competent SARS-CoV-2. (**F**). Percentage of viral replication in the presence of 20μM of camostat, cepharanthine, hydroxychloroquine, ivermectin and ciclesonide after infection with replication-competent SARS-CoV-2. Mean±SEM is shown. Data in panel 4F were analyzed by one sample t-test; *p<0.05, **p<0.01.

cell line. However, camostat was highly active in HLT cells ($EC_{50}$ = 3.3μM). However, in a clinical trial with COVID-19 patients receiving camostat within the first 48h of admission, no significant benefit was observed [49]. Yet, the potential benefit of camostat during the early phase of infection remains to be addressed. Further, valaciclovir, an antiviral drug, presented some inhibitory potential at 100 μM only in HLT cells. Interestingly, phenformin, an antidiabetic drug and an mTOR inhibitor, has been postulated as an inhaled drug candidate against influenza and coronavirus infections [50]. Phenformin reduced the incidence of influenza infection in diabetic patients during the 1971 outbreak [50]. Here, we detected that phenformin significantly reduced viral entry at 20–100μM only in HLT cells, supporting previous recommendations as inhaled treatment. Finally, eriodictyol, a flavonoid used as a medicinal plant [51], demonstrated certain activity starting at 4μM. In contrast, quercetin induced some viral entry suppression only in VeroE6 cells and at high concentrations (**Figs 4A–4C**). To further demonstrate the suitability of our model to test antivirals against other SARS-CoV-2 variants, we tested the inhibitory capacity of the drugs with the lowest $IC_{50}$ values, cepharanthine, camostat, ivermectin, hydroxychloroquine, and ciclesonide using a pseudovirus containing the SARS-CoV-2 spike of the Delta variant. Similar inhibitory dynamics for all tested drugs were shown using both viral variants, indicating the versatility of our model (**Fig 4D**).

Last, we verified our main findings in HLT cells using a replication-competent SARS-CoV-2 virus. Although we initially also included ergoloid, we observed high toxicity in HLT cells (>60% cell toxicity) at longer incubation times (48h), which precluded the execution of the antiviral assay (**S4F Fig**). HLT cells were infected with a SARS-CoV-2 clinical isolate and viral genomes were measured in the supernatant of cell cultures by RT-PCR in the presence of the different drugs. SARS-CoV-2 successfully replicated in HLT samples (**Fig 4E**), and viral replication was partially inhibited by all five drugs (**Fig 4F**). These results are consistent with previous reports describing the antiviral potency of camostat using precision-cut lung slices from donors [48,52]. Overall, these results indicate that HLT cells represent a reproducible and relevant system for the screening of antivirals in a physiological model. This system not only recapitulates the main antiviral activities observed in cell models, but also allows the identification of new compounds missed by conventional systems.

## Impact of inflammation and anti-inflammatory drugs on ACE2 expression and SARS-CoV-2 viral entry

Since SARS-CoV-2 viral infection rapidly induces an inflammatory response, we wondered if certain components of this response could modulate ACE2 expression, potentially increasing viral binding of SARS-CoV-2 and thus, enhancing infection. Further, ACE2 has been previously identified as an ISG or a component of the IFN-signaling pathway [13,53], and a recent investigation showed that cultured human primary basal epithelial cells treated with IFN-α2 and IFN-γ led to upregulation of *ACE2* [13]. Moreover, IL-1β and IFN-β upregulated ACE2 in large airway epithelial cell cultures [10]. Thus, considering that type I interferons represent a first line of defence against viral infections and that several cytokines are rapidly induced and associated with disease severity in COVID-19 patients [54], we tested the effects of different molecules on ACE2 expression in HLT cells. In initial experiments, we tested three different doses of a wider range of molecules (including tumor necrosis factor (TNF), IL-6 and IFN-γ that were subsequently discarded), which were used to select doses and compounds of interest. Finally, the effect of IFN-α2, IFN-β1, IL-1β, IL-10 and GM-CSF on ACE2 expression was evaluated in HLT cells. Cells were then treated with selected immune stimuli and cultured for 20h, when the expression of ACE2 in enriched AT-II cells was evaluated by flow cytometry. The only significant change we observed was for IL-1β stimulation, which decreased the fraction of

AT-II cells expressing ACE2 (**Fig 5A**). No other significant changes were observed, indicating that relevant inflammatory stimuli, besides IL-1β, have a limited impact on ACE2 expression in AT-II cells.

Moreover, it is currently not well documented if anti-inflammatory drugs could modulate ACE2 expression, and consequently, impact susceptibility to SARS-CoV-2 infection [55]. Several glucocorticoids have shown to impart activating effects on ACE2 expression in cell lines; cortisol showed the strongest effect on ACE2 activation, followed by prednisolone, dexamethasone, and methylprednisolone [17]. Moreover, NSAIDs, compounds that inhibit cyclooxygenase-1 and 2 mediating the production of prostaglandins, which play a role in inflammatory responses, have been linked to ACE2 upregulation [17]. Here, we use HLT cells to study the effect of several anti-inflammatory drugs on both ACE2 expression and SARS-CoV-2 viral entry. 1/5 dilutions of ibuprofen, cortisol, dexamethasone and prednisone were added to HLT cells for 20h. Overall, no effect on ACE2 expression was observed after the addition of these anti-inflammatory drugs (**Fig 5B**). Consequently, tested anti-inflammatory compounds showed no major impact of the viral entry assay; however, high concentrations of prednisone and dexamethasone showed a partial reduction of viral entry in HLT cells, without any apparent impact on VeroE6 cells (**Fig 5C**). Thus, selected anti-inflammatory drugs had limited impact on ACE2 expression within enriched AT-II cells from the HLT model, as well as in SARS-CoV-2 viral entry.

## Anti-inflammatory properties of selected compounds

Last, we were interested in modelling the anti-inflammatory properties of several drugs in HLT cells. Based on their previous antiviral potency, we selected camostat, cepharanthine, ciclesonide, ergoloid, hydroxychloroquine and ivermectin for further evaluation. Of note, some of these drugs have been previously identified as immunomodulators with anti-inflammatory effects (**S1 Text**). However, their direct impact on inflammatory molecules directly secreted by human lung cells have not been evaluated. HLT cells were stimulated with lipopolysaccharides (LPS) and IFN-γ in the presence of these antivirals and, 20h after, the expression of IL-6 and CXCL10, a potent pro-inflammatory cytokine and a chemokine respectively, were intracellularly measured by flow cytometry. IL-6 and CXCL10 were selected as molecules significantly increased in severe patients during acute infection, with a prediction value for hospitalization [56]. A representative flow cytometry gating strategy is shown in **S5 Fig**. As shown in **Fig 6A,** two major subpopulations of myeloid cells contributed to the upregulation of CXCL10 and IL-6 expression upon stimulation. Myeloid CD11b$^+$CD14$^+$ were the cells with a greater response, with 50% of these cells expressing IL-6 and 30% expressing CXCL10 after stimulation (**Fig 6B**). Using this model of local inflammation, we tested the capacity of the selected compounds to attenuate this response. We observed that camostat had the most potent effect, which significantly reduced the expression of CXCL10 in CD11b$^+$CD14$^-$ and of IL-6 in CD11b$^+$CD14$^+$ myeloid subsets (**Fig 6B**). Ergoloid, which has not been linked to modulation of inflammation before, significantly reduced the expression of cytokines in CD11b$^+$CD14$^+$ myeloid cells, and cepharanthine reduced IL-6 production within the same subset. In contrast, ciclesonide induced CXCL10 secretion in CD11b$^+$CD14$^-$ myeloid cells (**Fig 6B**). Altogether, our results validate the use of HLT cells as a relevant method for the identification of anti-inflammatory compounds impacting specific pro-inflammatory cell populations located in the lung parenchyma.

## Discussion

The emergency created by the fast spread of SARS-CoV-2 infection worldwide required a quick response from physicians treating these patients, who adapted to the rapid knowledge

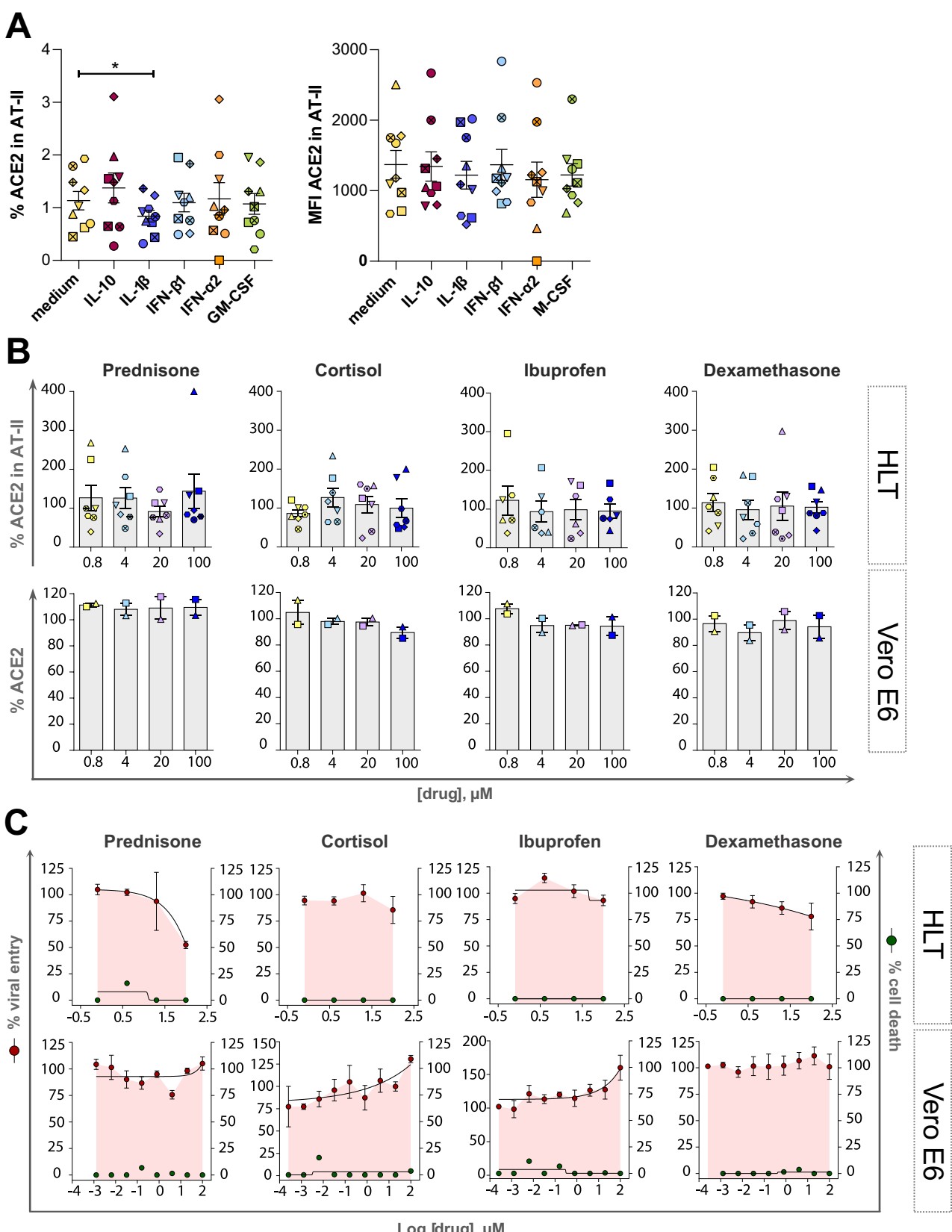

**Fig 5. Impact of inflammation and anti-inflammatory drugs on SARS-CoV-2 viral entry and ACE2 expression.** Both models, HLT cells and VeroE6 cells, were incubated in the presence of different anti-inflammatory drugs to evaluate the modulation of ACE2 expression by flow cytometry and the antiviral effect by luminescence. (**A**). HLT cells were treated with different stimuli for 20h and the percentage of protein expression (left) or the mean fluorescence intensity (MFI, right) of ACE2 receptor was evaluated in the enriched AT-II fraction by flow cytometry. (**B**) Modulation of ACE2 protein expression was assessed by flow cytometry in both models, Vero E6 and HLT cells, in the presence of different concentrations of each anti-inflammatory drug, ranging from 100μM to 0.8μM. Percentage of ACE2 expression was quantified in AT-II cells from at least six independent lung samples, and in VeroE6 cells from 2 independent experiments. (**C**). Viral entry on Vero E6 and HLT cells exposed to VSV*ΔG(Luc)-Spike in the presence of different concentrations of the anti-inflammatory drugs prednisone, cortisol, ibuprofen and dexamethasone. Drugs were used at a concentration ranging from 100μM to 0.256 nM, in VeroE6, and to 0.8μM in lung cells. Non-linear fit with variable response curve from at least two experiments in replicates is shown (red lines). Cytotoxic effect on Vero E6 cells and HLT cells exposed to different concentrations of drugs in the absence of virus is also shown (green lines). Mean±SEM are shown and statistical comparisons with the control medium were performed using the Wilcoxon test. *p<0.05.

being generated by both clinical practice and basic research. However, up to date, only one antiviral drug against SARS-CoV-2 has been approved for clinical use. New antivirals are urgently needed, and the choice of the cell and animal models used to test the efficacy of drugs will impact its rapid translation into the clinics. Here, we propose the use of human lung tissue (HLT) cells as a method that can be safely performed in a BSL2 facility, which allows i) the identification of cell targets and expression of viral entry factors, ii) the study of the impact of inflammation on host-pathogen interactions and iii) a rapid medium-high throughput drug screening of entry inhibitors against SARS-CoV-2 variants and local anti-inflammatory candidates.

Using pseudotyped viral particles expressing the SARS-CoV-2 spike, we first corroborated that a fraction of CD45- CD31- HLA-DR+ and EpCAM+ is enriched in AT-II cells and are the primary cell target in lung tissue in steady conditions. This agrees with several studies using different approximations [13,37,57] and validates our primary model for viral tropism identification. While cell lines have been traditionally used for the screening of potential antiviral compounds due to their reproducibility, as well as being quick and user-friendly assays, they lack physiological relevance. Similarly, entry receptors and viral factors have been identified using immortalized cell lines [12,58], and cell targets for SARS-CoV-2 in tissues have been mainly determined by analyzing the expression of viral entry factors in RNA-seq datasets [59] or using replication-competent SARS-CoV-2 isolates in BSL3 facilities [60]. Importantly, these studies have identified AT-II cells as main viral targets for SARS-CoV-2 infection in the lungs, and the molecules ACE2, CD147, TMPRSS2 and AXL as important factors for viral entry [12,35,38,61]. However, the development of more refined and translational *ex vivo* models of SARS-CoV-2 entry will not only have implications for understanding viral pathogenesis, but also will be useful for the characterization of cell targets under specific conditions or for the identification of potential antivirals blocking viral entry in primary cells. The direct use of HLT cells allows the maintenance of cell type diversity and it may represent a significant advantage over previous models [19,21,62].

Moreover, we showed that the HLT cells can be successfully used for drug screening purposes, not only against the D614G virus but also against other viral variants such as the Delta. We tested 39 drugs and compared the results with antiviral testing in VeroE6 cells. Not surprisingly, we showed discordant results between both methods. Indeed, we found that 33.3% of the tested compounds had discordant results between HLT and VeroE6 cells; 26.66% of drugs showed some antiviral effect in HLT but no activity was detected in VeroE6, and 6.67% showed only antiviral effects in VeroE6 cells. Among other reasons, the differential expression of several key proteins needed for viral entry, might explain current discrepancies between cell types. Importantly, using HLT cells, we identified several compounds with antiviral activity; cepharanthine showed an EC$_{50}$ of 6.08μM and concordantly, it was recently identified in a high throughput screening as one of the most potent drugs against SARS-CoV-2 [63], likewise several other studies have pointed towards this drug as a potent entry and post-entry

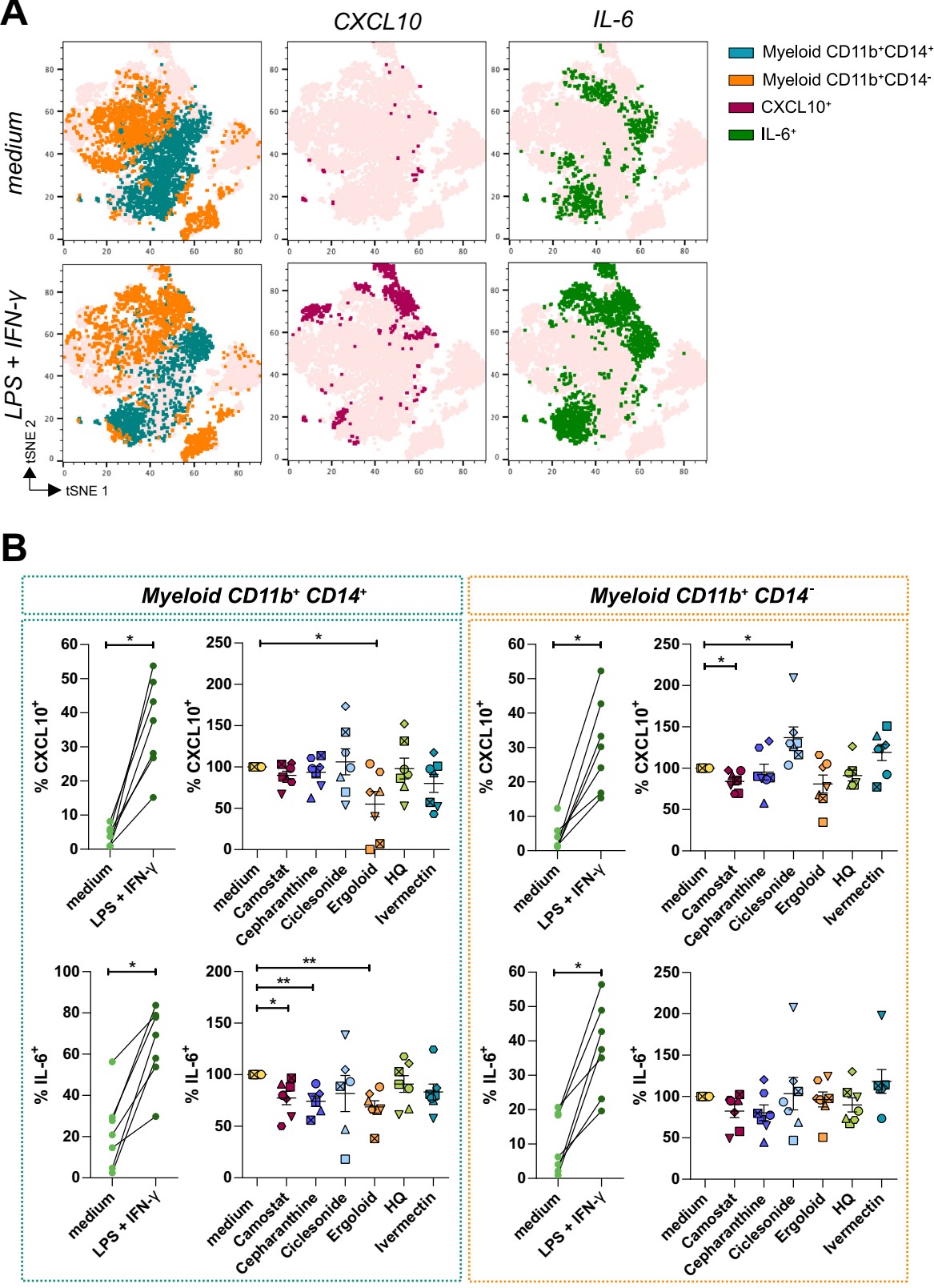

**Fig 6. Anti-inflammatory effect of compounds with antiviral activity against SARS-CoV-2.** HLT cells were cultured in the presence of 20μM of camostat, cepharanthine, ciclesonide, ergoloid, hydroxychloroquine (HQ) and ivermectin, alone or in combination with the stimuli

LPS (50 ng/ml) and IFN-γ (100 ng/ml). (**A**) *t*-distributed Stochastic Neighbor Embedding (t-SNE) representations displaying the major cell clusters present in live CD45[+] myeloid gate, based on FSC and SSC, of a representative human lung tissue in baseline conditions and after stimulation with LPS and IFN-γ. Two major subsets of myeloid cells are shown (CD11b[+] CD14[+], in blue-green, and CD11b[+] CD14[-], in orange). The expression of CXCL10 and IL-6 among the different populations is shown in maroon and green, respectively. (**B**) Expression of CXCL10 and IL-6 was measured in HLT cells in response to stimuli in the presence of selected drugs in both myeloid subpopulations, CD11b[+] CD14[+] (left panel) and CD11b[+] CD14[-] (right panel). Mean±SEM are represented and statistical comparisons with the control medium were performed using the One sample t test. [*]p<0.05, [**]p<0.01.

SARS-CoV-2 inhibitor [64]. Instead, for hydroxychloroquine, an early report suggested no antiviral activity in human lung cells due to different expression of the required proteases for viral entry [65]. Furthermore, clinical trials failed to show effectiveness of this drug as a treatment for COVID-19 [66–68]. A strong dependency of SARS-CoV-2 on TMPRSS2 for viral entry, rather than on cathepsin L, was identified as a possible mechanistic explanation for its failure *in vivo* [69]. In our study, however, we observed that this drug was equally effective at inhibiting viral entry in VeroE6 and HLT cells, and also was effective when using replication competent viral isolates in HLT cells. Concordantly, in differentiated air-liquid interface cultures of proximal airway epithelium and 3D organoid cultures of alveolar epithelium, hydroxychloroquine significantly reduced viral replication [70]. The multiple mechanisms of action postulated for hydroxychloroquine, including interference in the endocytic pathway, blockade of sialic acid receptors and restriction of pH mediated S protein cleavage at the ACE2 binding site [71], could help to explain its antiviral effect in primary lung cells. Similarly, we identified ivermectin as an effective antiviral in HLT cells. Of note, ivermectin received limited attention as a potential drug to be repurposed against COVID-19 based on its limited ability to reach lung tissue *in vivo* [72]. Further, a clinical trial failed to show a reduction in the proportion of PCR-positive patients seven days after ivermectin treatment [73].

Importantly, HLT cells also provide a platform for testing anti-inflammatory drugs and the modulation of viral entry factors with drug candidates and immunomodulatory stimuli. We showed that IL-1β was able to reduce ACE2 expression in the fraction of enriched AT-II cells, in contrast to other cytokines induced during SARS-CoV-2 infection like IFN-α2, IFN-β1, IL-10 and GM-CSF, which did not impact ACE2 protein production. In primary epithelial cells derived from healthy nasal mucosa, Ziegler et al. [13] showed a significant induction of ACE2 transcripts after IFN-α2 and IFN-γ stimulation, as well as and in a human bronchial cell line treated with either type I or type II IFN. Moreover, the authors showed that influenza A virus infection increased ACE2 expression in lung resections [13], strongly suggesting that ACE2 was an ISG. However, following studies showed that ACE2 transcription and protein production was not responsive to IFN. Instead, they described a new RNA isoform, MIRb-ACE2, that was highly responsive to IFN stimulation, but importantly, encoded a truncated and unstable protein product [74,75]. These results highlight the need to validate scRNA-seq data with orthogonal approaches, such as the confirmation of protein expression levels in relevant systems. In HLT cells, we quantified ACE2 protein expression and importantly, focused our analysis on putative AT-II cells, the main SARS-CoV-2 targets in lung parenchyma. Also, in agreement with our results, in a primary human bronchial epithelial cell model, type I (β), II (γ), or III (λ1) IFNs did not induce ACE2 expression [76]. Moreover, a study performed by Lang et al [77], showed that IFN-γ and IL-4 downregulate the SARS-CoV receptor ACE2 in VeroE6 cells, and similarly, stimulation of A549 cells with IFN-α, IFN-γ, and IFN-α+IFN-γ did not identify *ACE2* as an ISG [78].

A feasible explanation for the decrease of ACE2 protein production upon IL-1β treatment is that IL-1β activates disintegrin and metalloproteinase domain-containing protein 17 (ADAM17) [79], which mediates the shedding of ACE2 [80]. Although this effect would seem

positive to reduce SARS-CoV-2 infection, ACE2 is a lung-protective factor, as it converts Angiotensin (Ang) II to Ang-(1–7); while Ang II promotes harmful effects in the lung, e. g. fibrosis, vasoconstriction, inflammation, endothelial dysfunction, edema, and neutrophil accumulation [81], Ang-(1–7) has counter-regulatory effects protective of lung injury. Moreover, Ang-(1–7) plays an essential role in hemostasis, as it favors anti-thrombotic activity in platelets [82]. In any case, treatment of COVID-19 patients with respiratory insufficiency and hyper inflammation with IL-1 inhibitors was associated with a significant reduction of mortality [83], indicating that at least during severe COVID-19 the overall effect of IL-1β is detrimental. While the reduction of ACE2 expression in AT-II cells by IL-1β may be of interest, it needs to be determined if in combination with other cytokines rapidly induced during viral respiratory infection [84], this effect would remain. Further, glucocorticoids and NSAIDS have been linked to ACE2 upregulation previously [17]. In contrast, we did not observe any significant impact of ibuprofen, cortisol, dexamethasone and prednisone on ACE2 protein expression. These results are concordant with a recent report showing that suppression of cyclooxygenase (COX)-2 by two commonly used NSAIDs, ibuprofen and meloxicam, had no effect on ACE2 expression, viral entry, or viral replication in a mouse model of SARS-CoV-2 infection [85]. Moreover, dexamethasone incompletely reduced viral entry. This observation partially agrees with a study using lung cells previously treated with dexamethasone, which showed significant suppression of SARS-CoV-2 viral growth [26].

We additionally show the interest of the HLT model to test local inflammation and evaluate potential anti-inflammatory drugs. The culture of diverse cell subsets localized in the lung parenchyma, without further cell separation, allows the detection of inflammatory responses generated by different resident subpopulations, which is a significant advantage over monoculture. Several resident-myeloid subsets, together with newly recruited ones, may contribute to the rapid cytokine storm detected in COVID-19 patients [86–88]. Thus, the identification of antiviral drugs that can also limit the extent of these initial pro-inflammatory events may offer added value to the overall therapeutic effect of a given drug. In this sense, we observed that camostat significantly reduced the expression of proinflammatory molecules IL-6 and CXCL10 in several myeloid CD11b[+] subsets. Concordantly, in a previous study using primary cultures of human tracheal epithelial cells infected with H1N1 virus, camostat also reduced the concentrations of the cytokines IL-6 and TNF in cell supernatants [89], suggesting a potent anti-inflammatory potential. In contrast, ivermectin did not affect the expression of cytokines in our model. However, ivermectin was previously shown to have protective anti-inflammatory effects in mice, reducing the production of TNF, IL-1 and IL-6 *in vivo* and *in vitro* [90]. Of note, in our system we optimized the detection of changes for the intracellular expression of IL-6 and CXCL10 by local myeloid cells, and thus, how these intracellular changes reflect total production in supernatant needs further evaluation.

Finally, it is also important to note the potential limitations of the model. First, we did not maintain the cells in an air-liquid interface, which may alter cell function. Other limitations include the limited availability of human lung samples, inter-patient variation (age, smoking, etc.), the effects on lung biology of the medical condition instigating surgery, and the exact location of the sample resection, which may affect the proportion of cell subsets such as AT-II. However, this variability is what shapes the HLT into a more physiological and relevant model in comparison to current methods based on immortalized cell cultures. Besides the interest of the different readouts from HLT cell-based system as proposed here, our results highlight drugs with antiviral activity together with immunomodulatory properties, which could increase the benefit of a given treatment during COVID-19 disease progression. For instance, camostat, cepharanthine and ergoloid were three of the most potent drugs inhibiting SARS-CoV-2 entry, and remarkably, also exerted a significant anti-inflammatory effect on myeloid

cells. Clinical trials with camostat, ergoloid and cepharanthine, ideally administrated during early infection, should shed light on their use as both antivirals and anti-inflammatory compounds.

## Materials and methods

### Ethics statement

Study protocol was approved by the Comitè d'Ètica d'Investigació Clínica (Institutional Review Board number PR(AG)212/2020) of the Hospital Universitari Vall d'Hebron in Barcelona, Spain. Samples were obtained from adults, all of whom provided written informed consent.

### Cells and virus

VeroE6, isolated from kidney epithelial cells of an African green monkey, were grown in DMEM medium supplemented with 10% fetal bovine serum (FBS; Gibco) 100 U/ml penicillin, and 100 μg/ml streptomycin (Capricorn Scientific) (D10) and maintained at 37˚C in a 5% $CO_2$ incubator.

The spike of the SARS-CoV-2 virus (D614G variant) was generated (GeneArt Gene Synthesis, ThermoFisher Scientific) from the codon-optimized sequence obtained by Ou et al. [42] and inserted into pcDNA3.1D/V5-His-TOPO (pcDNA3.1-S-CoV2Δ19-G614). The spike of the SARS-CoV-2.SctΔ19 B.1.617.2 (Delta) virus was generated (GeneArt Gene Synthesis, ThermoFisher Scientific) from the full protein sequence of the original SARS-CoV-2 isolate Wuhan-Hu-1 (WH1) modified to include the mutations specific to the Delta variant (VOC-21APR-02: T19R, 157-158del, L452R, T478K, D614G, P681R, D950N). These plasmids present the mutation D614G and a deletion in the last 19 amino acids from the original spike. Pseudo-typed viral stocks of VSV*ΔG(Luc)-S were generated following the protocol described by Whitt [91] with some modifications. Briefly, 293T cells were transfected with 3μg of the plasmid encoding the SARS-CoV-2 spike. Next day, cells were infected with a VSV-G-Luc virus (MOI = 1) (generated from a lentiviral backbone plasmid that uses a VSV promoter to express luciferase) for 2h and gently washed with PBS. Cells were incubated overnight in D10 supplemented with 10% of I1 hybridoma (anti-VSV-G) supernatant (ATCC CRL-2700) to neutralize contaminating VSV*ΔG(Luc)-G particles. Next day, the resulting viral particles were collected and titrated in VeroE6 cells by enzyme luminescence assay (Britelite plus kit; PerkinElmer), as described previously [92].

### Lung tissue

Lung tissues were obtained from patients without previous COVID-19 history and a recent negative PCR test for SARS-CoV-2 infection undergoing thoracic surgical resection at the Thoracic Surgery Service of the Vall d'Hebron University Hospital. Non-neoplastic tissue areas were collected in antibiotic-containing RPMI 1640 and immediately dissected into approximately 8-mm$^3$ blocks. These blocks were first enzymatically digested with 5 mg/ml collagenase IV (Gibco) and 100 μg/ml of DNase I (Roche) for 30 min at 37˚C and 400 rpm and, then, mechanically digested with a pestle. The resulting cellular suspension was filtered through a 70μm-pore size cell strainer (Labclinics) and washed twice with PBS. Pellet recovered after centrifugation was resuspended in fresh medium (RPMI 1640 supplemented with 5% FBS, 100 U/ml penicillin, and 100 μg/ml streptomycin) and DNase I to dissolve cell aggregates, and the resulting cell suspension was then filtered through a 40μm-pore size cell strainer (Labclinics). Cell number and viability were assessed with LUNA Automated Cell Counter

(Logos Biosystems). For cell phenotyping the following antibodies were used: anti-CD31 (PerCP-Cy5.5, BioLegend), anti-CD11b (FITC, BioLegend), anti-CD11c (PE-Cy7, BD Biosciences), anti-E-cadherin (PE-CF594, BD Biosciences), primary goat anti-ACE2 (R&D systems), anti-CD14 (APC-H7, BD Biosciences), anti-CD45 (AF700, BioLegend), anti-EpCAM (APC, BioLegend), anti-CD3 (BV650, BD Biosciences), anti-CD15 (BV605, BD Biosciences) and anti-HLA-DR (BV421, BioLegend). For ACE2 detection, after surface staining, cells were stained with secondary donkey anti-goat IgG (PE, R&D Systems) for 30 min at 4˚C. Cell viability was determined using an AQUA viability dye for flow cytometry (LIVE/DEAD fixable AQUA, Invitrogen). In some experiments, instead of CD11b or CD15, we used a primary rabbit anti-TMPRSS2 or anti-CD147 (BV605, BD Biosciences), respectively. For TMPRSS2 detection, after ACE2 staining with the appropriate secondary antibody, cells were washed twice with PBS 1% NMS (normal mouse serum) and then stained with a secondary goat anti-rabbit IgG (AF488, Thermofisher) for 30 min at 4˚C. For SPC detection, after surface staining with a primary rabbit anti-SPC antibody (Biorbyt) and instead of ACE2 staining, cells were stained with a secondary donkey anti-rabbit IgG (PE, Biolegend) for 30 min at 4˚C. After fixation with PBS 2% PFA, cells were acquired in an LSR Fortessa (BD Biosciences), and data were analyzed using the FlowJo v10.6.1 software (TreeStar).

## Cytospin and alkaline phosphatase staining

Cytospin preparations were obtained from freshly isolated human lung cells at an approximate density of 150,000 cells/slide, and air-dried during 15 min. Cells were stained with alkaline phosphatase, as an enzyme marking epithelial type II cells, following manufacturer's instructions (Alkaline phosphatase Kit, Sigma). The intensity of pink stain reflects the amount of alkaline phosphatase in positive cells.

## ACE2 immunohistochemical staining in human lung tissue sections

Human lungs were maintained in 10% formalin for 24 hours and then embedded in paraffin. Paraffin-embedded lungs were cut into 4 μm sections. After removing the paraffin, endogenous peroxidases were inactivated in an aqueous solution containing 3% $H_2O_2$ and 10% methanol and antigen retrieval was performed heating the samples in citrate buffer (10mM citric acid, pH 6.0). The sections were then blocked in bovine serum albumin (5%), incubated with anti-ACE2 antibody (R&D Systems cat. n˚ AF933, dilution 1:100) and with biotinylated secondary antibody against goat IgGs (Vector Laboratories cat. n˚ BA-9500, dilution 1:250). Proteins were visualized using the ABC Peroxidase Standard Staining Kit (ThermoFisher) followed by 3,3′-Diaminobenzidine (DAB) Enhanced Liquid Substrate System (Sigma Aldrich). Counterstaining was done with hematoxylin.

## Antiviral screening assay

The complete list of compounds tested in this study, including information about its clinical use, product reference and vendors is shown in **S1 Text**. Duplicates of five-fold serial dilutions of 39 antiviral compounds were tested in both VeroE6 cell line and in human lung tissue (HLT) cells using at least 2 different donors. For VeroE6, five-fold serial dilutions of the compounds, ranging from 100μM to 0.25nM, were prepared in D10 in 96-well flat-bottom plates. VeroE6 cells were added at a density of 30.000 cells/well and incubated with the drug for at least 1 h before infection. Subsequently, cells were infected with 1,500 $TCID_{50}$ of VSV*ΔG (Luc)-S virus. In parallel, drug cytotoxicity was monitored by luminescence. To evaluate the antiviral activity of drugs in HLT cells, five-fold serial dilutions of the compounds, ranging from 100μM to 0.8μM or 6.4nM, were prepared in R10 in 96-well conic-bottom plates. HLT

cells were added at a density of 300,000 cells/well and incubated with the compound for at least 1h before infection. Then, MOI 0.1 of VSV*ΔG(Luc)-S virus were added to wells, and plates were spinoculated at 1,200g and 37˚C for 2h. After infection, fresh medium was added to the wells and cell suspensions were transferred into a 96-well flat-bottom plate. Cells were then cultured overnight at 37˚C in a 5% $CO_2$ incubator. Each plate contained the following controls: no cells (background control), cells treated with medium (mock infection) and cells infected but untreated (infection control). After 20h, cells were incubated with Britelite plus reagent (Britelite plus kit; PerkinElmer) and then transferred to an opaque black plate. Luminescence was immediately recorded by a luminescence plate reader (LUMIstar Omega). To evaluate cytotoxicity, we used the CellTiter-Glo Luminescent kit (Promega), following the manufacturer's instructions. Data was normalized to the mock-infected control, after which $EC_{50}$ and $CC_{50}$ values were calculated using Graph-Pad Prism 7.

## Drug validation with replication competent SARS-CoV-2

These experiments were performed in BSL3 facilities (Viral Vector Production Unit, Universitat Autònoma de Barcelona, UAB; and the Centre de Medicina Comparativa i Bioimatge de Catalunya (CMCiB)). The SARS-CoV-2 virus was isolated from a nasopharyngeal swab from an infected patient hospitalized at the Vall d'Hebron Hospital. VeroE6 cells were cultured on a cell culture flask (25 $cm^2$) at $1.5 \times 10^6$ cells overnight prior to inoculation with 1 mL of medium from a Deltaswab VICUM tub containing the swab. Cells were cultured for 1h at 37˚C and 5% $CO_2$. Afterwards, DMEM containing 2% FCS were added to the cells and incubated for 48h. Cells were assessed daily for cytopathic effect and the supernatant was recollected and subjected to viral titration in VeroE6 by plaque assay.

For antiviral drug validation, HLT samples were incubated with different drugs at 20μM for at least 1h before infection. Tested drugs were camostat, cepharanthine, ergoloid, hydroxychloroquine, ivermectin and ciclesonide. Then, cells were infected with a MOI 0.5 of the SARS-CoV-2 viral isolate, and the plate was incubated for 2h at 37˚C and 5% CO2. After infection, samples were extensively washed with PBS 1X to eliminate residual virus and suspended in fresh media containing antiviral drugs and transferred into a new plate. 24 or 48h post infection, 140μl of supernatant was collected in tubes containing 140μl of DNA/RNA Shield (Zymo Research) for SARS-CoV-2 inactivation. For each experiment, a negative control, cells treated with only medium, and a positive control, cells incubated in the presence of the virus alone, were included. Percentage of viral infection was calculated by RT-PCR. Briefly, viral RNA from the supernatant was extracted using the QIAamp Viral RNA Mini Kit (Qiagen), following the manufacturer's instructions. RNA was reverse transcribed with SuperScript III (Invitrogen), in accordance with the instructions provided by the manufacturer, and cDNA was quantified by qPCR using the 2019-nCoV CDC RUO Kit (IDT, catalog #10006713) for the detection of viral RNA of the nucleocapsid region N1 from the SARS-CoV-2 (N1 forward 5′-GACCCCAAAATCAGCGAAAT-3′ and N1 reverse 5′-TCTGGTTACTGCCAGTTGAA TCTG-3′; N1 probe 5′-FAMACCCCGCAT/ZEN/TACGTTT GGTGGACC-3IABkFQ-3′). Copies of SARS-CoV-2 RNA were quantified using a standard (2019-nCoV_N_Positive Control from IDT, catalog #10006625). Samples were run on a 7000 SDS instrument (Applied Biosystems).

## Modulation of ACE2 expression by anti-inflammatory drugs and immune stimuli

VeroE6 and lung cells were incubated with five-fold serial dilutions of selected anti-inflammatory compounds (ranging from 100μM to 0.8μM) for 20h. Tested drugs included cortisol,

ibuprofen, prednisone and dexamethasone. Lung cells were also incubated with the following cytokines: GM-CSF (100 ng/ml, Immunotools), IL-1β (10 ng/ml, Immunotools), IL-10 (100 ng/ml, Immunotools), IFN-β (100 U/ml, Immunotools), or IFN-α2 (100 U/ml, Sigma Aldrich). For determination of ACE2 expression, the following surface staining antibodies were used: primary goat anti-ACE2 (R&D Systems), anti-CD45 (AF700, BioLegend), anti-EpCAM (APC, BioLegend), and anti-HLA-DR (BV421, BioLegend). For ACE2 detection, cells were then stained with secondary donkey anti-goat IgG (PE, R&D Systems) for 30 min at 4˚C. A Fluorescent Minus One control (FMO) without primary anti-ACE2 antibody was used as a control. Cell viability was determined using an AQUA viability dye for flow cytometry (LIVE/ DEAD fixable AQUA, Invitrogen). After fixation with PBS 2% PFA, cells were acquired in an LSR Fortessa (BD Biosciences) and analyzed using the FlowJo v10.6.1 software (TreeStar).

## Immunomodulatory capacity of selected drugs

HLT cells were cultured in a round-bottom 96-well plate containing 20 μM of cepharanthine, ergoloid, ciclesonide, hydroxychloroquine, ivermectin, or camostat mesylate alone or in combination with the stimuli LPS (50 ng/ml) and IFN-γ (100 ng/ml). For each patient, a negative control, cells treated with only medium, and a positive control, cells incubated in the presence of LPS and IFN-γ, were included. Immediately, brefeldin A (BD Biosciences) and monensin (BD Biosciences) were added to cells and cultured overnight at 37˚C in 5% $CO_2$. Next day, cellular suspensions were stained with the following antibodies: anti-CD11b (FITC, BioLegend), anti-CD69 (PE-CF594, BD Biosciences), anti-CD14 (APC-H7, BD Biosciences), anti-EpCAM (APC, BioLegend), anti-CD3 (BV650, BD Biosciences), anti-CD45 (BV605, BioLegend), and anti-HLA-DR (BV421, BioLegend). Cells were subsequently fixed and permeabilized using the Cytofix/Cytoperm kit (BD Biosciences) and intracellularly stained with anti-IL-6 (PE-Cy7, BioLegend), and anti-CXCL10 (PE, BioLegend). Cell viability was determined using an AQUA viability dye for flow cytometry (LIVE/DEAD fixable AQUA, Invitrogen). After fixation with PBS 2% PFA, cells were acquired in an LSR Fortessa (BD Biosciences), and data were analyzed using the FlowJo v10.6.1 software (TreeStar).

## Statistical analyses

Statistical analyses were performed with Prism software, version 6.0 (GraphPad). A P value <0.05 was considered significant.

## Supporting information

**S1 Fig. Visual summary of the HLT model.** *Ex vivo* physiological systems for the study of SARS-CoV-2-host interactions are scarce. We establish a method using primary human lung tissue (HLT) cells for the rapid analysis of cell tropism and identification of therapeutics. Main findings: **i)** HLT cells preserve main cell subpopulations, including alveolar type-II cells, and expression of SARS-CoV-2 entry factors ACE2, CD147, TMPRSS2 and AXL. **ii)** HLT cells are readily susceptible to SARS-CoV-2 infection without the need of cell isolation or further cell differentiation. **iii)** Antiviral testing in HLT cells allows the rapid identification of new drug candidates against SARS-CoV-2 variants, missed by conventional systems. **iv)** Local inflammation is supported in HLT cells and offers the identification of relevant anti-inflammatory compounds for SARS-CoV-2 infection.
(EPS)

**S2 Fig. Gating strategy for the identification of cell subpopulations in the human lung tissue model.** (**A**) General gating strategy used to identify different cell subsets in lung samples.

A gate based on FSC vs. SSC was followed by doublet and dead cells exclusion. From live CD45⁻ cells, endothelial cells (CD31⁺, purple) and epithelial cells (EpCAM⁺, grey) were gated, and within epithelial cells, AT-II cells (EpCAM⁺ and HLA-DR⁺, pink) were identified. Out of live CD45⁺ cells and based on FSC vs. SSC, we identified a lymphocyte population in which we distinguished between non-T lymphocytes (turquoise) and T cells (dark green) based on CD3 expression; and big cells, where we identified three major subsets based on their expression of CD11b and CD11c and, subsequently, CD14 and HLA-DR markers. We identified alveolar macrophages (blue), monocytes (violet), myeloid dendritic cells (mDCs, fuchsia) and neutrophils (orange).
(EPS)

**S3 Fig. Optimization of lung tissue enzymatic digestion visualized by *t*-distributed Stochastic Neighbor Embedding (tSNE), and representative SPC and ACE2 expression.** (**A**) Representative tSNE maps showing concatenated flow cytometry standard files for three different protocols based on different digestion enzymes (collagenase, liberase or trypsin) from total live cells (upper), CD45⁺ cells (middle) and CD45⁻ cells (lower). (**B**) Bar plots showing cell-type composition (count) analyzed by flow cytometry for each tissue protocol. (**C**) Representative flow cytometry plots showing Surfactant Protein C (SPC) staining and its respective fluorescence minus one (FMO) control. (**D**) Representative flow cytometry plots showing ACE2 staining and its respective fluorescence minus one (FMO) control.
(EPS)

**S4 Fig.** (**A**) Percentage of enriched AT-II cells co-expressing the entry factors ACE2 and CD147 (in blue), and ACE2 and TMPRSS2 (in purple). (**B**) *t*-distributed Stochastic Neighbor Embedding (tSNE) representation for AXL expression in CD45⁺ and CD45⁻EpCAM⁺ fractions from a representative lung tissue. Right graphs show the percentage of expression of the AXL entry factor in the different cell populations, which were identified as in Fig 1A. (**C**) Bar plots showing the percentage of viral entry inhibition on HLT cells in the presence of anti-ACE2 antibody (15μg/ml) or recombinant human AXL (50μg/ml) after cell challenge with VSV*ΔG (Luc)-Spike. Data were analyzed by one sample t-test; $^*$p<0.05, $^{**}$p<0.01. (**D**) Frequency of each subset relative to live cells at 0h and 24h with and without the presence of virus. All cell subsets were identified as shown in S2A Fig. (**E**) EC$_{50}$ values in the HLT model obtained from 3 different lung donors and performed in replicates. (**F**) Cells from 1 donor were cultured with 20 μM of selected drugs for 48h, and cell toxicity was measured using the CellTiter-Glo Luminescent kit (Promega), following the manufacturer's instructions. Data was normalized to the untreated control. Mean±SEM is shown for all graphs.
(EPS)

**S5 Fig. Gating strategy for the identification of anti-inflammatory effects of selected compounds.** General gating strategy used to evaluate the expression of inflammatory molecules in lung samples. A gate based on FSC vs. SSC was followed by doublet and dead cells exclusion. From live CD45⁺ cells and based on FSC vs. SSC, we identified lymphocyte population and big cells, in which we identified two subsets based on their expression of CD11b and CD14: myeloid CD11b⁺CD14⁺ cells (blue-green) and myeloid CD11b⁺CD14⁻ cells (orange) are shown.
(EPS)

**S1 Text. Antiviral drug candidates for entry inhibition of SARS-CoV-2.**
(DOCX)

**S2 Text. EC$_{50}$ and CC$_{50}$ of 39 antiviral drug candidates.**
(DOCX)

## Author Contributions

**Conceptualization:** Meritxell Genescà, Maria J. Buzon.

**Data curation:** Judith Grau-Expósito, Meritxell Genescà, Maria J. Buzon.

**Formal analysis:** Judith Grau-Expósito, David Perea, Marina Suppi, Meritxell Genescà, Maria J. Buzon.

**Funding acquisition:** Meritxell Genescà, Maria J. Buzon.

**Investigation:** Meritxell Genescà, Maria J. Buzon.

**Methodology:** Judith Grau-Expósito, David Perea, Marina Suppi, Núria Massana, Ander Vergara, Maria José Soler, Benjamin Trinite, Julià Blanco, Javier García-Pérez, José Alcamí, Anna Serrano-Mollar, Joel Rosado, Vicenç Falcó.

**Project administration:** Meritxell Genescà, Maria J. Buzon.

**Resources:** Benjamin Trinite, Julià Blanco, Javier García-Pérez, José Alcamí, Joel Rosado.

**Supervision:** Meritxell Genescà, Maria J. Buzon.

**Visualization:** Judith Grau-Expósito, David Perea, Marina Suppi.

**Writing – original draft:** Judith Grau-Expósito, Maria J. Buzon.

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
