## [Decision Letter · Decision Letter 0]

6 Dec 2021

Dear Dr. Buzon,

We are pleased to inform you that your manuscript 'Evaluation of SARS-CoV-2 Entry, Inflammation and New Therapeutics in Human Lung Tissue Cells' has been provisionally accepted for publication in PLOS Pathogens.

Before your manuscript can be formally accepted we recommend you to address comments by reviewer # 3.  You will also need to complete some formatting changes, which you will receive in a follow up email. A member of our team will be in touch with a set of requests.

Best regards,

Rudra Channappanavar, DVM, Ph.D.

Guest Editor

PLOS Pathogens

Carolina Lopez

Section Editor

PLOS Pathogens

Kasturi Haldar

Editor-in-Chief

PLOS Pathogens

orcid.org/0000-0001-5065-158X

Michael Malim

Editor-in-Chief

PLOS Pathogens

orcid.org/0000-0002-7699-2064

Reviewer Comments (if any, and for reference):

Reviewer's Responses to Questions

**Part I - Summary**

Reviewer #1: The revised manuscript is satisfactory and I will recommend it to be accepted.

Reviewer #2: (No Response)

Reviewer #3: This manuscript evaluates SARS-CoV-2 entry and host response using a a novel human lung tissue model. Strength of this manuscript include, i) a reliable and novel model to rapidly assess the virus infection and therapeutics in human cells/tissue, ii) Offers benefit of assessing epithelial and immune cell cross talk during viral infection, ii) additional advantage of identifying antiviral effects of compounds shown to be not effective in traditional Vero cell culture system.

Weakness: some of the results presented are redundant as several previous studies showed similar outcome. Despite this weakness, utility of this model to asses host response and test therapeutic is of interest to broader scientific community.

**Part II – Major Issues: Key Experiments Required for Acceptance**

Reviewer #1: N/A

Reviewer #2: (No Response)

Reviewer #3: Although LPS+ IFNg stimulation model is a well established, evaluating inflammation (Figure 6) in HLT/immune cell model following SARS-CoV-2 infection would complement results presented in Figure 1-4. If CoV-2 infection data is available, this reviewer recommends authors to include it with current results.

**Part III – Minor Issues: Editorial and Data Presentation Modifications**

Reviewer #1: N/A

Reviewer #2: (No Response)

Reviewer #3: (No Response)

PLOS authors have the option to publish the peer review history of their article (what does this mean?). If published, this will include your full peer review and any attached files.

Reviewer #1: **Yes: **Jian Zheng

Reviewer #2: **Yes: **Jincun Zhao

Reviewer #3: No

---

## [Editor Report · Acceptance letter]

6 Jan 2022

Dear PhD Buzon,

We are delighted to inform you that your manuscript, "Evaluation of SARS-CoV-2 Entry, Inflammation and New Therapeutics in Human Lung Tissue Cells," has been formally accepted for publication in PLOS Pathogens.

Best regards,

Kasturi Haldar

Editor-in-Chief

PLOS Pathogens

orcid.org/0000-0001-5065-158X

Michael Malim

Editor-in-Chief

PLOS Pathogens

orcid.org/0000-0002-7699-2064